# Comparison of Cortisol, Androstenedione and Metanephrines to Assess Selectivity and Lateralization of Adrenal Vein Sampling in Primary Aldosteronism

**DOI:** 10.3390/jcm10204755

**Published:** 2021-10-17

**Authors:** Giulio Ceolotto, Giorgia Antonelli, Brasilina Caroccia, Michele Battistel, Giulio Barbiero, Mario Plebani, Gian Paolo Rossi

**Affiliations:** 1Specialized Center for Blood Pressure Disorders-Regione Veneto and Emergency Medicine-Hypertension Unit, Department of Medicine-DIMED, University of Padua, 35126 Padua, Italy; giulio.ceolotto@unipd.it (G.C.); brasilina.caroccia@unipd.it (B.C.); 2Laboratory Medicine, Department of Medicine-DIMED, University of Padua, 35126 Padua, Italy; giorgia.antonelli@unipd.it (G.A.); mario.plebani@unipd.it (M.P.); 3Institute of Radiology, Department of Medicine-DIMED, University of Padua, 35126 Padua, Italy; michele.battistel@aopd.veneto.it (M.B.); giulio.barbiero@aopd.veneto.it (G.B.)

**Keywords:** adrenal vein sampling, cortisol, androstenedione, metanephrines, primary aldosteronism

## Abstract

Success of adrenal vein sampling (AVS) is verified by the selectivity index (SI), i.e., by a step-up of cortisol levels between the adrenal vein and the infrarenal inferior vena cava samples, beyond a given cut-off. We tested the hypothesis that androstenedione, metanephrine, and normetanephrine, which have higher gradients than cortisol, could increase the rate of AVS studies judged to be bilaterally successful and usable for the clinical decision making. We prospectively compared within-patient, head-to-head, the selectivity index of androstenedione (SI_A_), metanephrine (SI_M_), and normetanephrine (SI_NM_), and cortisol (SI_C_) in consecutive hypertensive patients with primary aldosteronism submitted to AVS. Main outcome measures were rate of bilateral success, SI values, and identification of unilateral PA. We recruited 136 patients (55 + 10 years, 35% women). Compared to the SI_C_, the SI_A_ values were 3.5-fold higher bilaterally, and the SI_M_ values were 7-fold and 4.4-fold higher on the right and the left side, respectively. With the SI_A_ and the SI_M_ the rate of bilaterally successful AVS increased by 14% and 15%, respectively without impairing the identification of unilateral PA. We concluded that androstenedione and metanephrine outperformed cortisol for ascertaining AVS success, thus increasing the AVS studies useable for the clinical decision making.

## 1. Introduction

All available guidelines recommend adrenal vein sampling (AVS) for the identification of primary aldosteronism (PA) patients, who wish to achieve long-term surgical cure [1,2,3]. The clinical use of this technically challenging procedure requires the obtainment of blood draining from the adrenal gland, which is commonly verified by demonstrating a step-up above a given cut-off value of plasma cortisol levels between each of the adrenal veins and the infra-renal inferior vena cava, i.e., by calculating that the cortisol selectivity index (SI_C_) [4].

This seemingly straightforward approach is hampered in practice by the low step-up of cortisol, which can lead to judge as non-selective AVS studies that in reality are selective. We hypothesized that biomarkers with a higher step-up than cortisol can improve the rate of bilateral success of catheterization [5,6], a contention that, if verified, would be a major step forward in this field in that it could allow use of a higher proportion of AVS studies for the clinical decision making.

Of note, metanephrine (M) and normetanephrine (NM), the products of epinephrine and norepinephrine metabolism by COMT, were suggested to have a higher step-up between adrenal veins and inferior vena cava plasma than cortisol [5,6]. Likewise, 17α OH-progesterone, the product of 17 α-hydroxylase, and androstenedione, the precursor of adrenocortical androgens, albeit circulating at levels much lower than cortisol, were suggested to allow rescuing AVS studies, which were judged to be unsuccessful by using the SI_C_, for diagnostic purposes in a small pilot study [7]. Whether these biomarkers could consistently improve the assessment of AVS selectivity, and furnish a higher rate of bilaterally selective AVS studies over cortisol, and how they compared with metanephrine and normetanephrine remained, however, to be verified in prospective studies. Equally important, whether these biomarkers could allow to establish lateralization of PA as accurately as cortisol when used in calculation of the lateralization index remained altogether unknown. Therefore, in a sizable cohort of PA patients consecutively submitted to AVS at our institution, we prospectively compared within-patients the selectivity index of androstenedione (SI_A_), metanephrine (SI_M_), normetanephrine (SI_NM_), and cortisol (SI_C_) and tested the hypotheses that use of these selectivity biomarkers could increase the rate bilaterally successful AVS studies without impairing the assessment of lateralization, as compared to cortisol, for identification of unilateral surgically cured PA.

## 2. Materials and Methods

### 2.1. Patients

Consecutive hypertensive patients with biochemically confirmed PA, who wished to pursue surgical cure, were prospectively recruited for the AVS study at University of Padua, an ESH Center of Excellence, if they had no contraindications to surgery and/or general anesthesia following current guidelines [1,2]. Imaging of the adrenal glands with contrast medium computed tomography was performed in all patients to assess venous drainage [1,2]. The patients were prepared for AVS by switching treatment to a long-acting calcium channel blocker and/or doxazosin and after attainment of normokalemia, according to guidelines [1,8]. Unilateral surgically curable PA was determined by demonstrating biochemical cure of PA, i.e., the normalization of plasma aldosterone concentration (PAC) and serum K levels at one month and 6 months follow-up after unilateral adrenalectomy.

Procedures followed the Helsinki Declaration Principles; all patients provided informed written consent and the protocol was approved by the Ethics Committee of the University of Padua.

### 2.2. Adrenal Vein Sampling

AVS was performed with the bilaterally simultaneous technique by highly experienced radiologists (MB and GB) using catheters shaped for each adrenal vein [8]. Blood was slowly collected in a lithium heparin BD Vacutainer^®^ PST™ Tube by gravity from the right and left adrenal veins and from the infrarenal inferior vena cava for the measurements of plasma aldosterone concentration (PAC), plasma cortisol concentration (PCC), androstenedione (A), normetanephrine (NM) and metanephrine (M). The samples were immediately refrigerated and centrifuged at 4000 g for 5 min at 4° C, divided into plasma aliquots (600 μL), and immediately frozen and stored at −80 °C for analysis. The selectivity index for cortisol (SI_C_), androstenedione (SI_A_), metanephrine (SI_M_) and normetanephrine (SI_NM_) was calculated as the ratio between the level of each analyte measured in each adrenal vein and in the infrarenal inferior vena cava, as originally described for cortisol [4].

### 2.3. Analytical Methods

All measurements were performed in an ISO 9001 and ISO 15189-certified laboratory by technicians, blind to the clinical diagnosis and the imaging data, under strict supervision of experienced Laboratory Medicine phySI_C_ians (GA and MP). PAC was measured using the prospectively validated [9] commercially available chemiluminescent assay LIAISON XLTM Aldosterone kit (Diasorin). Cortisol and androstenedione were determined as described [7].

Metanephrine and normetanephrine levels were measured by a CE-IVD kit for liquid chromatography tandem mass spectrometry (LC-MS/MS) determination (81000 Chromsystems^®^, Grafelfing, Germany). Samples were prepared following the manufacturer instructions. Briefly, 500 μL of dilution buffer, 25 μL of internal standard mix solution and 500 μL of plasma sample or calibrator or control were applied to a 96-solid phase extraction (SPE) well plate. After three washing steps, M and NM were eluted with 250 μL of elution buffer on the collection plate. The collection plate was sealed and placed in the LC–MS/MS autosampler at 8 °C. The analytes separation was performed using an analytical column kept at a constant temperature of 28 °C and an appropriate gradient elution profile; quantification was carried out on LCMS-8060 system (Shimadzu Corporation, Tokyo, Japan), with electrospray ionization source. The volume injection was 10 μL and the total run time was 5 min. The optimized MS conditions were as follows: interface temperature 300 °C, desolvation line temperature 300 °C, heat block temperature 450 °C, nebulizing gas flow 3 L/min, heating gas flow 14 L/min, drying gas flow 5 L/min. Quantitative analysis was performed in multiple reaction monitoring (MRM) mode. Two positive ion selected transitions were monitored for M and NM, and a single ion transition was monitored for d3 M and d3 NM, using the optimized parameters.

Six-point calibration curves were obtained by plotting the ratios between the analyte peak area and internal standard peak area against concentration in pmol/L. The analytical measurement ranges were 116–13,749 pmol/L and 117–23,277 pmol/L for M and NM, respectively. The validation of each analysis batch was made with three internal quality controls-IQCs (Chromsystems^®^, Grafelfing, Germany; target values: L1 267 pmol/L, L2 859 pmol/L, L3 5010 pmol/L for M; L1 650 pmol/L, L2 1393 pmol/L, L3 7395 pmol/L for NM). The analytical batch was validated if all IQCs levels were within ±15% of the target values. Samples with M or NM levels above the upper limit of linearity (28·6 nmol/L for M and 46·4 nmol/L for NM, as stated in the IFU) were diluted with isotonic saline solution, as recommended in the assay datasheet.

The imprecision monitored with the three ICQs (*n* = 23) demonstrated a coefficient of variation (CV%) of 6% and 7% at L1, 2% and 5% at L2 and 2% and 4% at L3, for M and NM respectively.

### 2.4. Statistical Methods

Within-patient and between-group comparison of quantitative variables was performed with paired and unpaired Student’s t-tests after a normal distribution was obtained with transformation of skewed variables. A non-parametric Wilcoxon’s test was used when transformation did not furnish a gaussian distribution. The categorical distribution of variables was compared by way of Chi-square analysis. Significance was set at *p* < 0.05. For statistical analysis we used the SPSS (version 27 for Mac, IBM, Italy), the GraphPad Prism (version 9.00 for Mac, GraphPad Software, La Jolla, CA, USA) and, for ROC construction and curves comparison the MedCalc (version 19.1.5) software was used.

## 3. Results

### 3.1. Clinical Features

The main clinical features of the PA patients recruited in this study divided according to their being assigned to adrenalectomy or medical therapy are shown in Table 1.

The adrenalectomized group showed higher PAC and ARR, and a trend toward higher systolic BP values at baseline. Biochemical cure, i.e., the normalization of PAC, ARR, and serum K^+^ values, a rise in plasma active renin concentration, along with a highly significant fall of BP, was achieved after surgery in all. The medically treated patients, owing to an intensification of the multiple drug treatment, showed a rise of renin, the normalization of high BP, along with correction of the hypokalemia. Concurrent excess cortisol secretion, i.e., the so-called “Conn-shing” syndrome [10], was not observed in any PA patients.

### 3.2. Hormonal Data

Table 2 shows the values of cortisol, androstenedione, metanephrines, and normetanephrine at the time of AVS. In the infrarenal IVC blood, taken as a surrogate of peripheral venous blood, androstenedione showed plasma values in the nanomolar range, i.e., on average were 100-fold lower than cortisol; the values of M and MN were also much lower (about 2.50 × 10^−3^ and 1.29 × 10^−3^ fold, respectively), than cortisol.

Notwithstanding the lower absolute values, the SI_A_ values were on average about 3.5-fold higher than the SI_C_ values. The values of the SI_M_ were 7-fold and 4.4-fold higher on the right and the left side, respectively, than the corresponding SI_C_.

The SI_NM_ were 0.94-fold higher than the SI_C_ on the right side, but lower (0.6-fold) on the left side.

The SI_A_ and SI_C_ values showed a tight correlation (*p* < 0.001), with a regression line shifted to higher values for the SI_A_ (Figure 1, Panels A and B). A significant correlation was also found between the SI_M_ and SI_NM_, between SI_M_ and SI_NM_ and SI for either steroid (Figure 1, Panels C to H).

For all the biomarkers the SI values were higher on the right than on the left side (Figure 2, Figure 3 and Figure 4).

### 3.3. Effect of Gender on the SI Values

As biomarkers like androstenedione, are known to show between genders differences, we investigated if there were sex differences of SI values, and if they affected the rate of bilateral selectivity. We found no evidence for a significant gender effect: in both men and women the SI_A_ and SI_M_, and the rate of bilateral selectivity with these SI, were consistently higher than the SI_C_, in keeping with what observed in the mixed gender cohort.

### 3.4. Effect of Timing of Blood Sampling on the SI Values

As our institutional protocol for AVS requires two sets of blood sampling, one immediately after catherization (t0) and one 15 min later (t15), we could assess the effect of timing of blood sampling for all four biomarkers. The comparison of SI individual values obtained at t0 and at t15 on both sides showed SI_A_ and SI_M_ values that were consistently and significantly higher than the SI_C_ at both time points (Figure 2, Figure 3 and Figure 4).

Notably, the SI values were higher at t0 than at t15 for both SI_C_ and SI_A_ values, but not for SI_M_ and SI_NM_ (Figure 5).

Consequently, using the ≥2.0 SI cut-off (recommended by experts for the for the SI_C_ under baseline (unstimulated) AVS studies [11]), the rate of bilaterally successful AVS studies was higher with the SI_C_, the SI_A_, and the SI_M_, but not for the SI_NM_ at t0 than at t15 (Table 3). The systematic use of either androstenedione or metanephrine, as selectivity biomarker, rescued between 12% and 17% of the AVS that were judged not to be successful by using the SI_C_.

### 3.5. Accuracy of Identification of Unilateral Surgically Cured PA by Using Androstenedione and Metanephrines

Biochemical cure after unilateral adrenalectomy was used to assess the accuracy of the lateralization index calculated by using androstenedione, metanephrine, and normetanephrine, as compared to cortisol.

The accuracy (area under the receiver operator characteristics (ROC) curve) of the lateralization index calculated with the different biomarkers (Table 4 and Figure 6) was high for all biomarkers and showed no significant differences across biomarkers (Appendix A).

## 4. Discussion

A recent large survey of AVS performance carried out in four continents showed that by using cortisol to determine the success of adrenal vein catheterization, one out of five AVS studies were judged to be unsuccessful, i.e., not bilaterally selective, and thus discarded from the clinical use [12], which means that the patients were considered ineligible for unilateral adrenalectomy, and thus denied the chance of long-term cure. This was a disconcerting performance for a procedure that is time-consuming, expensive, and invasive, as these patients did not benefit from being submitted to AVS.

In the AVIS-2 Selectivity study, use of more liberal SI cut-off values enlarged the proportion of patients judged to have a successful AVS study, who could be allocated to surgery with no evidence for worsened clinical outcomes after surgery [12]. An additional strategy for circumventing the low success rate of AVS could be the intra-procedural rapid cortisol assay, which is currently being tested in an ongoing randomized prospective study [13].

We herein investigated the performance of biomarkers that could perform better than cortisol for assessing selectivity, including androstenedione, metanephrine, and normetanephrine. We found that androstenedione and metanephrine displayed a higher concentration gradient between the adrenal vein and the inferior vena cava blood, in line with previous studies [6,7,14,15].

Of note, the levels of these biomarkers in plasma were much lower than those of cortisol (Table 2); hence, it is likely that their higher step-ups are accounted for by a slower clearance from the circulation rather than a higher rate of secretion by the adrenals. However, regardless of the underlying mechanism(s), it is worth pointing out that between 12% and 17%, of the AVS studies considered to be unsuccessful by the SI_C_ were judged to selective, i.e., successful (Table 3) by using the SI cut-off ≥2·0 recommended for the SI_C_ [11].

The finding of higher SI values for both androstenedione and metanephrine than for cortisol extended previous results [6,7,14,16], in that they showed a consistently better performance over the SI_C_ for ascertaining bilateral selectivity in our mixed gender cohort. Of note, this was not the case for the SI_NM_ (Table 2), likely because the adrenal medulla releases epinephrine and its product metanephrine at a larger amount than norepinephrine and its metabolite normetanephrine.

We regard these findings as important new pieces of information, because androstenedione can be accurately measured in most laboratories with a commercial kit with no need for the expensive instrumental set-ups used in previous studies [5,14]. In our hands, plasma metanephrine could also be quantified with a commercially available kit; however, technical expertise was necessary to optimize the liquid chromatography tandem mass spectrometry (LC-MS/MS) method and control for possible interferences [17].

The selectivity index of cortisol and androstenedione, and of normetanephrine and metanephrine showed tight highly significant direct correlations (Figure 1A,B), while less tight, albeit yet significant correlations, were seen also between the SI of either steroids and the metanephrines. This was expected given that catechol-O-methyl transferase (COMT), the enzyme metabolizing catecholamines to metanephrines, is regulated by adrenocortical hormones [18]. Moreover, the direct correlation between SI_C_ and SI_M_ found under unstimulated conditions immediately after the catheterization (at t0) and 15 min later (at t15) is consistent with results of a study where the majority of the 86 patients were investigated after cosyntropin stimulation [6]. Of note, for both SI_A_ and SI_M_ the regression line was shifted to the left indicating that both biomarkers provided higher SI values than cortisol (Figure 1A–D).

This study provided several further interesting observations. The first was that all the SI values were higher on the right than on the left side, probably due to the anatomical differences between the right and left adrenal vein drainage. Whether different cut-off values should be used to assess selectivity on each side, remains to be further investigated with outcome-based validation studies.

The second important novel information relates to the gender differences: the SI for androstenedione and metanephrine was higher than the SI_C_ in both men and women, as it was in the entire study cohort. To the best of our knowledge, this is the first study to explore SI differences according to gender.

The third important finding asks for a standardization of the AVS procedure: the values of the SI for cortisol and androstenedione were higher in blood samples obtained immediately after the catheterization than 15 min later, likely because of the waning of a reaction to the stress and pain associated with starting the procedure, and the ensuing decrease of the ACTH drive to steroid hormone secretion [19]. This time effect was not observed for metanephrine and normetanephrine, which showed a slight increase for reasons that remain to be clarified, as these catecholamine metabolites are constitutively released from the adrenal medulla. Of note, the higher values of SI_C_, and SI_A_ when starting catherization was associated with a slight, but significant, increase of the proportion of studies judged to be selective (Figure 5 and Table 3). As AVS studies performed under cosyntropin stimulation [20] have shown that the stress-induced adrenocortical hormone stimulation has an adverse impact on lateralization in many patients, the beneficial effect of stress on selectivity could be counterbalanced by a less accurate detection of lateralization. An unsolved issue is, therefore, whether to prefer samples taken immediately after catheterization that are associated with a higher rate of bilateral success when using the SI_C_ (Table 3), or to use samples taken 15 min afterwards, when the stress reaction might have waned, the rate of bilateral success could be lower, but there might be no untoward effect on the lateralization index. Interestingly, we found that for SI_A_, and SI_M_, the rate of bilateral success was only slightly higher at t0 than at t15. Thus, with these biomarkers the waning of the stress reaction did not significantly worsen the assessment of AVS success.

An even clinically more important novel finding of this study entails the similarly high accuracy for the identification of unilateral surgically cured PA of the different biomarkers, when used in the calculation of the lateralization index at ROC curve analysis (Table 4 and Figure 6), indicating that they can effectively replace cortisol to this aim. However, it should be noted that as compared to cortisol, the use of androstenedione and metanephrine extended by about 13% and 15%, respectively, the size of the cohort where the lateralization index could be clinically used, because the AVS was judged to be bilaterally selective.

The yield of androstenedione and metanephrines in the PA patients with concurrent excess cortisol and aldosterone secretion, i.e., the so-called “Conn-shing” syndrome, who likely comprise a tiny proportion of the cases, could not be established in this study, because we did not observe any such cases. Therefore, this is another issue that needs further investigation, because thus far only a case report suggested the usefulness of using metanephrine to identify lateralization in Conn-shing syndrome [16].

In summary, by exploiting use of a head-to-head within-patient comparison, our results showed that both androstenedione and metanephrines measurements provided unambiguously higher SI values, and a higher rate of bilaterally selective studies than cortisol, thus improving the success of AVS. Importantly, even though they increase the number of clinically usable AVS studies, they did not worsen the diagnosis of unilateral surgically cured PA. Therefore, we would like to contend that they should replace cortisol for use in the calculation of AVS results.

In perspective, the obvious most clinically important issue, which is worth of further investigative research, is whether the replacement of cortisol with these biomarkers in the calculation of the lateralization index can translate into a better clinical outcome. Addressing this issue will require a study where patients will be randomized to treatment based on the different SI values using clinical outcome as a reference.

## Figures and Tables

**Figure 1 jcm-10-04755-f001:**
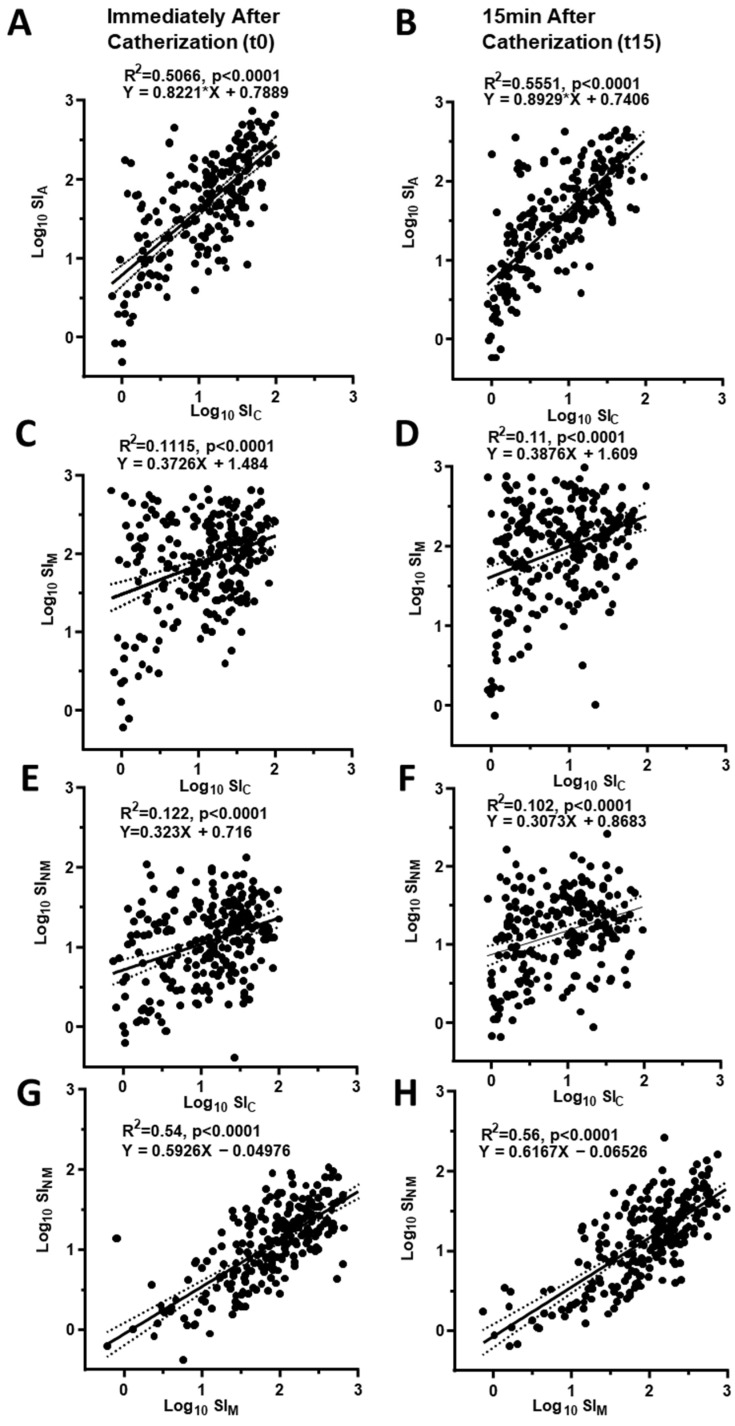
The scatter plots show the relationship between the log10-transformed values of the selectivity index calculated from cortisol (SI_C_) and androstenedione (SI_A_) immediately after starting catherization (Panel **A**) and after 15 min (Panel **B**). Similar plots illustrate the correlation of SI_C_ with SI calculated from metanephrine (SI_M_, Panels **C** and **D**) and normetanephrine (SI_NM_, Panels **E** and **F**). The relationship between SI_M_ and SI_NM_ is also shown (Panels **G** and **H**). Please note that the correlation between the steroids (Panels **A** and **B**), and between the metanephrines (Panels **C** and **H**) was tighter than between steroids ad metanephrines.

**Figure 2 jcm-10-04755-f002:**
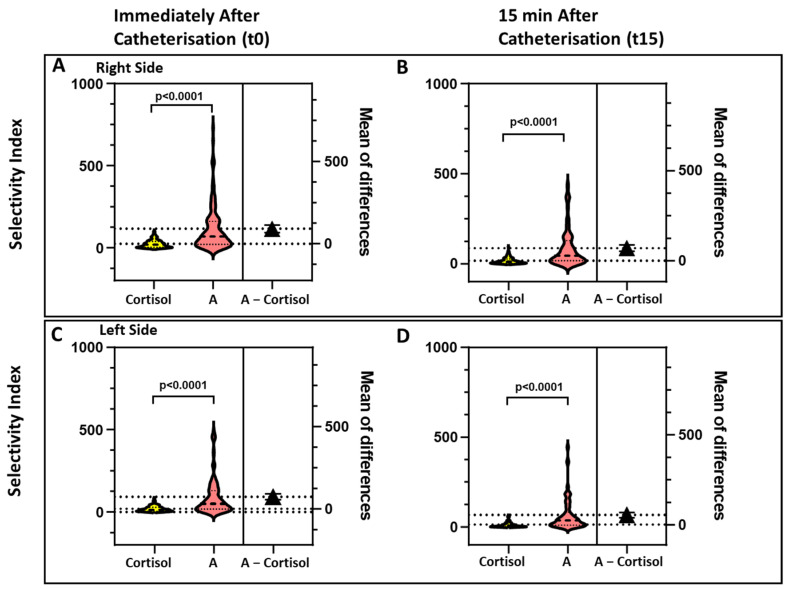
The violin plots show the values of the selectivity index for cortisol (SI_C_) and androstenedione (SI_A_), measured at t0, i.e., immediately after starting catherization (Panels **A** and **C**), and 15 min afterward (t15, Panels **B** and **D**), on the right side (Panels **A** and **B)** and left side (Panels **C** and **D**). For both sides and time points the SI_A_ was bilaterally higher than the SI_C_ in a highly significant manner.

**Figure 3 jcm-10-04755-f003:**
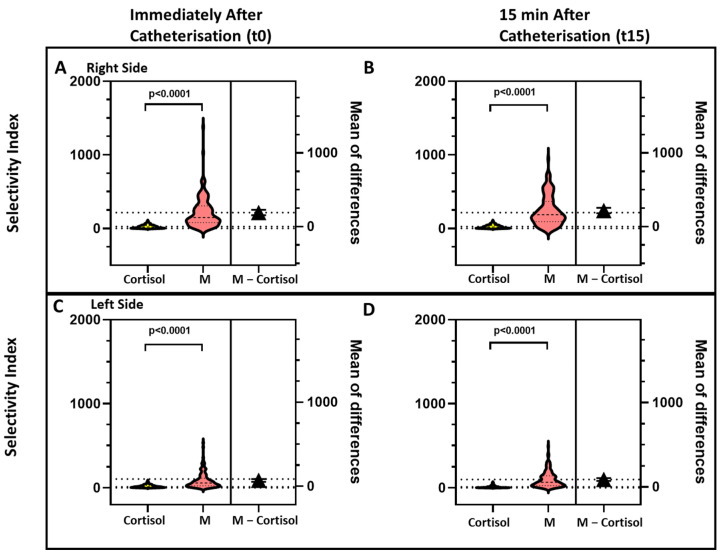
The violin plots show the values of the selectivity index for cortisol (SI_C_) and metanephrine (SI_M_), measured immediately after starting catheterization (Panels **A** and **C**) and 15 min afterward (Panels **B** and **D**), on the right side (Panels **A** and **B**) and left side (Panels **C** and **D**). For both sides and time points the SI_M_ was higher than the SI_C_ in a highly significant manner on both sides.

**Figure 4 jcm-10-04755-f004:**
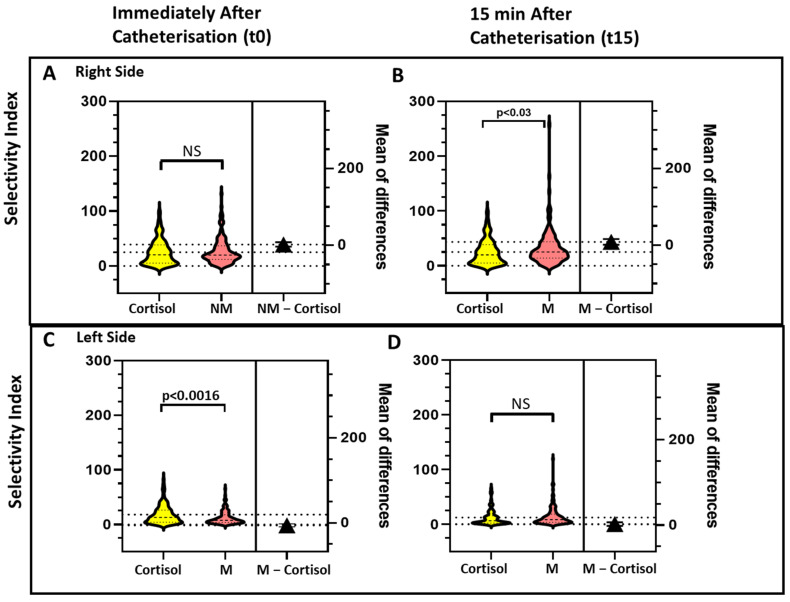
The violin plots show the values of the selectivity index for cortisol (SI_C_) and for normetanephrine (SI_NM_), measured immediately after starting catherization (Panels **A** and **C**) and 15 min afterward (Panels **B** and **D**), on the right side (Panels **A** and **B**) and left side (Panels **C** and **D**). The SI_NM_ was higher at baseline than 15 afterward in a highly significant manner on both sides.

**Figure 5 jcm-10-04755-f005:**
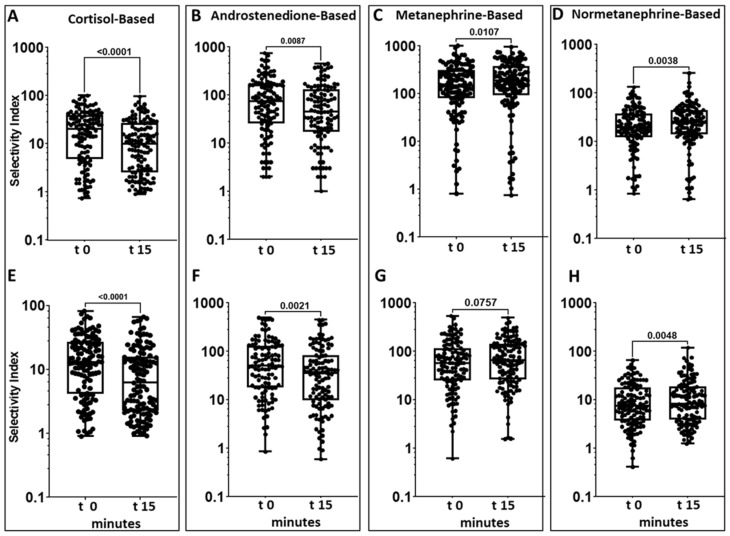
The box and whiskers plots show that on both the right (Panels **A** and **B**) and the left side (Panels **E** and **F**) the SI values for cortisol and androstenedione were higher when starting catheterization (t-15) than 15 min afterwards (t0). The SI for metanephrine and normetanephrine showed and opposite trend (Panels **C** and **D** for right side, Panel **G** and **H** for left side), likely because of the constitutive adrenomedullary release of these biomarkers.

**Figure 6 jcm-10-04755-f006:**
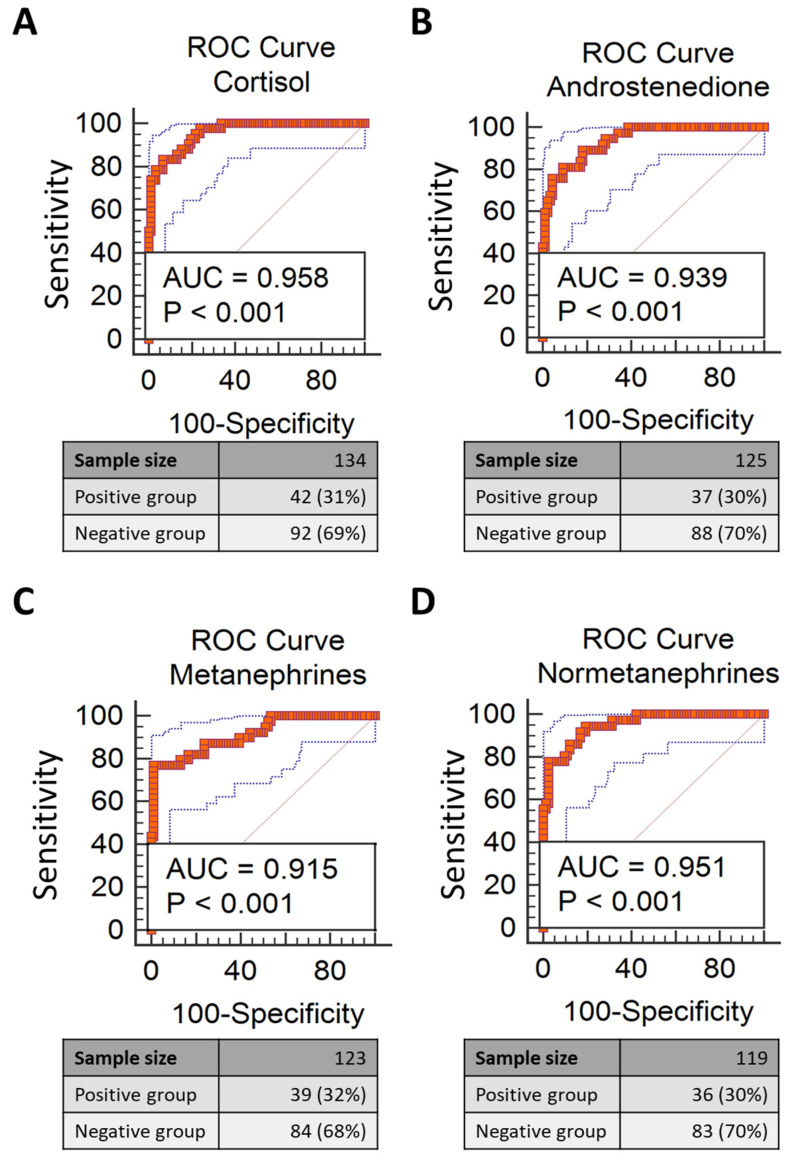
Receiver operating characteristic (ROC) curve of the lateralization index calculated using cortisol (**A**), androstenedione (**B**), metanephrine (**C**) and normetanephrine (**D**) for the identification of unilateral surgically cured PA. Dotted lines represent the confidence bands.

**Table 1 jcm-10-04755-t001:** Demographic and clinical features at baseline and after surgical or medical-treatment of the PA patients.

Variable	Surgically Treated (*n* = 79)			Medically-Treated (*n* = 57)		
	Before	After	*p*	Baseline	Follow-up	*p*
Age (years)	54 ± 10			56 ± 12		
Gender (% F)	36%			35%		
BMI (Kg/m^2^)	26.0 ± 3.4	25.4 ± 4.0	0.65	26.5 ± 3.1	27.2 ± 2.0	0.45
BSA (m^2^)	1.91 ± 0.02	1.90 ± 0.3	0.67	1.96 ± 0.2	1.91 ± 0.2	0.87
Systolic BP (mmHg)	150 ± 18*	133 ± 11	10^−3^	143 ± 17	139 ± 10	0.35
Diastolic BP (mmHg)	90 ± 13	80 ± 9	10^−3^	88 ± 9	82 ± 6	0.76
Serum Creatinine (mmol/L)	76 (64–83)	77 (66–83)	0.35	78 (70–84)	75 (68–81)	0.88
Serum Na^+^ (mmol/L)	140 ± 2	140 ± 2	0.10	140 ± 3	140 ± 1	0.4
Serum K^+^ (mmol/L)	3.4 ± 0.5	4.3 ± 0.8	0.001	3.6 ± 0.4	4.3 ± 0.5	0.19
DRC (mIU/L)	2.0 (2.0–3.5) *	3.5 (2.4–13)	0.001	2.5 (2.0–10)	2.8 (2–7.9)	0.34
PAC (ng/dL)	21.5 (14–40) **	5.1 (3.2–6.9)	0.001	12.7 (9–20)	18 (11–26)	0.67
ARR (ng/mIU)	88.1 (55–160) **	8.6 (3.9–14)	0.001	38.7 (15–67)	48 (20–82)	0.87

BP= blood pressure; BMI = body mass index; DRC = direct renin concentration; PAC = plasma aldosterone concentration; ARR = aldosterone-renin ratio. Data are presented as mean ± SD or median (IQR), as appropriate. * *p* < 0.05; ** *p* < 0.01 before vs medically-treated at baseline.

**Table 2 jcm-10-04755-t002:** Plasma concentrations and selectivity index values at t0 in adrenal veins and IVC for cortisol (SI_C_), androstenedione (SI_A_), metanephrine (SI_M_) and normetanephrine (SI_NM_).

Variable	Value		Value
Cortisol	nmol/L	Selectivity Index	
Infrarenal IVC	449 (335–576)		
Right adrenal vein	8829 (1731–20,200)	Right SI_C_	21.0 (5.2–39.4)
Left adrenal vein	5904 (1607–13,299)	Left SI_C_	13.1 (4.3–26.3)
Androstenedione	nmol/L		
Infrarenal IVC	4.5 (3.4–6.1)		
Right adrenal vein	342.0 (70.2–648)	Right SI_A_	69.6 (19.7–160.3)
Left adrenal vein	240.1 (80.8–600.8)	Left SI_A_	49.2 (18.0–122.6)
Metanephrine	pmol/L		
Infrarenal IVC	175.5 (123–317)		
Right adrenal vein	32,450 (19,375–52,511)	Right SI_M_	148.3 (80–307)
Left adrenal vein	11,645 (4615–21,558)	Left SI_M_	57.4 (25–115)
Normetanephrine	pmol/L		
Infrarenal IVC	350.5 (236–505)		
Right adrenal vein	7058 (4568–12,087)	Right SI_NM_	19.7 (12–37)
Left adrenal vein	2817 (1442–5278)	Left SI_NM_	8.3 (4–14)

Data are presented as median and IQ range. IVC, peripheral cava vein; SI_A_ = selectivity index for androstenedione; SI_C_ = selectivity index for cortisol; SI_M_ = selectivity index for metanephrines; SI_NM_ = selectivity index for normetanephrine.

**Table 3 jcm-10-04755-t003:** Proportion of bilaterally successful AVS with SI cutoff ≥ 2.0.

Parameter	t0 min	t15 min	χ^2^
**SI_C_**	82.4%	71.6%	0.0001
**SI_A_**	95.2%	89.6%	0.0001
**SI_M_**	97.6%	94.3%	0.009
**SI_NM_**	84.0%	86.4%	0.0001

SI_C_: cortisol selectivity index; SI_A_: androstenedione selectivity index; SI_M_: metanephrine selectivity index; SI_NM_: normetanephrine selectivity index.

**Table 4 jcm-10-04755-t004:** Area under the receiver operating characteristic curves (ROC) with 95% confidence interval of the lateralization index used to diagnose a unilateral surgically cured form of primary aldosteronism calculated using cortisol, androstenedione, metanephrine and normetanephrine.

Variable	PA Patients (Unilateral/ Bilateral, *n*)	AUC	C.I.	*p* Value vs. Identity Line AUC
Cortisol	134 (42/92)	0.958	0.928–0.980	0.0001
Androstenedione	125 (37/88)	0.939	0.898–0.979	0.0001
Metanephrine	123 (39/84)	0.915	0.863–0.976	0.0001
Normetanephrine	119 (36/83)	0.951	0.914–0.988	0.0001

AUC, area under curve, C.I. 95% confidence Interval.

## Data Availability

The data presented in this study are available on request from the corresponding author.

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
