# Peer review of "Comparison of Cortisol, Androstenedione and Metanephrines to Assess Selectivity and Lateralization of Adrenal Vein Sampling in Primary Aldosteronism"

_jcm, 2021, doi:10.3390/jcm10204755_

Round 1

Reviewer 1 Report

The experiment is well-designed, and the manuscript is written in a comprehensive way to demonstrate the hypotheses followed by clear and step-by-step interpretation of results.

The major strength of the study is that it owns increased case number than other cohorts and proceeds with within-patient and head-to-head comparison of the 3 potential biomarker candidates androstenedione, metanephrine, and normetanephrine, compared with standard biomarker cortisol, which previous studies seldom compared these parameters at the same time.

The statistical analyses were carefully planned and conducted, and the results were generally well reported. The results support the hypothesis that higher step-up in biomarkers associates higher selectivity indexes and improves rates of bilateral success of AVS.

Few points to comment:

  1. In Results, Hormonal data Table 2 and Figure 1, is the value of the four biomarkers the mean of t0 and t15 samples according to the institutional protocol?

  1. In Results, 3.3 Effect of gender on the SI values and Discussion (Line 302), is it possible that the author provides the data of SI between genders?

  1. In Figure 5 and Table 3 and Discussion (Line 312), decreased SIC and SIA was noticed from t0 to t15, while SIM and SINM showed an opposite trend. However, both SIA and SIM showed higher bilateral success of t0 than t15. Is there possible explanation of the difference?

  1. In Table 2 and Table 3. In Table 2 the data showed a slightly higher right SINM and lower left SINM compared with SIC as SIA and SIM both outperformed SIC. In Table 3 SIA, SIM, and SINM showed better successful AVS proportion t0 and t15. Is there possible explanation that a non-significantly higher SINM provides a better successful AVS proportion?

  1. Table 3, the footnotes, “SINM: metanephrine selectivity index” should be “SINM: normetanephrine selectivity index”.

Author Response

Reviewer 1

The experiment is well-designed, and the manuscript is written in a comprehensive way to demonstrate the hypotheses followed by clear and step-by-step interpretation of results. The major strength of the study is that it owns increased case number than other cohorts and proceeds with within-patient and head-to-head comparison of the 3 potential biomarker candidates androstenedione, metanephrine, and normetanephrine, compared with standard biomarker cortisol, which previous studies seldom compared these parameters at the same time. The statistical analyses were carefully planned and conducted, and the results were generally well reported. The results support the hypothesis that higher step-up in biomarkers associates higher selectivity indexes and improves rates of bilateral success of AVS.

We appreciated the careful assessment of our study and the helpful comments. Our replies in an itemize fashion is provided here below

  1. Results, Hormonal data Table 2 and Figure 1, is the value of the four biomarkers the mean of t0 and t15 samples according to the institutional protocol? RE: The values reported in a table 2 were calculated at t0. In figure 1 the scatterplots on the left show the value at T0 and those on the right panels show the value at the T 15. All are collected according to our institutional protocols.
  2. In Results, 3.3 Effect of gender on the SI values and Discussion (Line 302), is it possible that the author provides the data of SI between genders? RE:  Thanks for this suggestion. However, since as mentioned in the text there were no differences between men and women, after an in-house discussion, we thought that providing the data by gender will only complicate the text and thus the efficacy of the main message conveyed to the readers. Hence, we would prefer to keep the text as it is.
  3. In Figure 5 and Table 3 and Discussion (Line 312), decreased SIC and SIA was noticed from t0 to t15, while SIM and SINM showed an opposite trend. However, both SIA and SIM showed higher bilateral success of t0 than t15. Is there possible explanation of the difference? RE: Thanks raising this interesting point. The explanation that we can offer is that while cortisol and androstenedione are under ACTH control, metanephrine and normetanephrine are not as they constitutively released. This can account for the decrease of SIC and SIA. The reason why SIM and SINM slightly, albeit significantly increased, is unknown. This has been added in the discussion.
  4. In Table 2 and Table 3. In Table 2 the data showed a slightly higher right SINM and lower left SINM compared with SIC as SIA and SIM both outperformed SIC. In Table 3 SIA, SIM, and SINM showed better successful AVS proportion t0 and t15. Is there possible explanation that a non-significantly higher SINM provides a better successful AVS proportion? RE: This is an intriguing observation which it probably can be explained by the fact that even a non-significant increase in the absolute value can translate it into a number of studies being beyond the cut off value, and thus judge to be selective, when results are expressed in a categorical way, i.e. selective or non-selective.
  5. Table 3, the footnotes, “SINM: metanephrine selectivity index” should be “SINM: normetanephrine selectivity index”. RE: This mistake has been fixed. Thanks. In addition, we have fixed some typos and improved some awkward sentences.

Reviewer 2 Report

Comment to the authors

Ceolotto G et al. aimed to study how androstenedione and metanephrines could increase the rate of successful AVS (adrenal vein sampling). The topic is really interesting, but the manuscript needs some revisions before to be published. 

The study (particularly abstract and discussion) is too much focused on the impact of these hormones in the evaluation of selectivity of AVS, but I think that it could be even more important the role of these markers in the diagnosis of PA subtypes. I suggest to modify the paper according to this comment.

The authors confirmed in the present study that SI for metanephrine and androstenedione are higher than SI for cortisol during adrenal vein sampling. This information is already known and published in literature.

I suggest to simplify the title as following: “Comparison of Cortisol, Androstenedione and Metanephrines to assess selectivity and lateralization of Adrenal Vein Sampling in Primary Aldosteronism”.

The fact that the SI is higher for androstenedione and metanephrines that for cortisol does not necessarily means that the diagnostic accuracy is higher. Theoretically, a higher SI may be necessary for androstenedione to provide reliable and reproducible diagnoses of PA subtypes.

The observation that SI are higher in the right compared to the left is not novel and was reported by many authors. The fact that the SI on the left side is lower does not mean at all that a lower SI should be used.

Author Response

Reviewer 2

Ceolotto G et al. aimed to study how androstenedione and metanephrines could increase the rate of successful AVS (adrenal vein sampling). The topic is really interesting, but the manuscript needs some revisions before to be published. We appreciated the comments and constructive criticisms.

  1. The study (particularly abstract and discussion) is too much focused on the impact of these hormones in the evaluation of selectivity of AVS, but I think that it could be even more important the role of these markers in the diagnosis of PA subtypes. I suggest to modify the paper according to this comment. The authors confirmed in the present study that SI for metanephrine and androstenedione are higher than SI for cortisol during adrenal vein sampling. This information is already known and published in literature. RE: We appreciated the comments and constructive criticism and agree that the role of these markers could be even more important in the diagnosis of PA subtypes. However, based on the recent results of the AVIS-2 Study, where a fifth of the AVS studies had to be discarded from clinical use because they were judged to be non selective, we believe that the possibility of using a greater proportion of AVS studies for the clinical decision making that was documented in our study represents the most important finding of our study. Moreover, as the main effect of using these biomarkers was on increasing the number of selective studies and their use did not endanger the assessment of lateralization, we think that the main emphasis should be on judging success of AVS rather than on subtyping by means of the LI.
  2. I suggest to simplify the title as following: “Comparison of Cortisol, Androstenedione and Metanephrines to assess selectivity and lateralization of Adrenal Vein Sampling in Primary Aldosteronism”. RE: We agree with this statement and have changed the title as suggested.
  3. The fact that the SI is higher for androstenedione and metanephrines that for cortisol does not necessarily means that the diagnostic accuracy is higher. Theoretically, a higher SI may be necessary for androstenedione to provide reliable and reproducible diagnoses of PA subtypes. RE: Thanks for raising this point. We assume that the Reviewer meant a higher LI not the SI. When we started the study, we also thought that with better biomarkers the optimal LI cutoffs could differ. However, our data showed that this was not the case. In fact, the increase of the number of studies that could be used clinically, because they were considered to be bilaterally successful, did not result into the need of using higher cutoffs for assessing lateralization.
  4. The observation that SI are higher in the right compared to the left is not novel and was reported by many authors. The fact that the SI on the left side is lower does not mean at all that a lower SI should be used. RE: The statement that the higher SI on the right then on the left side was already known is probably true. However, we could not find it in the literature and would appreciate if the Reviewer could provide specific reference(s) that we would be glad to cite.

Round 2

Reviewer 2 Report

In my opinion, now the authors have modified the manuscript making it suitable for publication in JCM.